# Examining the Role of Life Skills in Mediating the Relationship between the Basic Psychological Needs and Subjective Well-Being of Taekwondo Student-Athletes

**DOI:** 10.3390/ijerph182111538

**Published:** 2021-11-02

**Authors:** Jun-Su Bae, Eun-Hyung Cho, Tae-Hee Lim

**Affiliations:** 1Department of Physical Education, Yongin University, Yongin 17092, Korea; 201971404@yiu.ac.kr; 2Korea Institute of Sport Science, Seoul 01794, Korea; ehcho@kspo.or.kr; 3Department of Taekwondo, Yongin University, Yongin 17092, Korea

**Keywords:** positive youth development, youth sport, psychosocial development, mediation effect

## Abstract

Various theories in the field of positive youth development (PYD) through sport argue that student athletes’ satisfaction with basic psychological needs, life skills development, and well-being are closely related to each other. This study identified the structural relationship among three basic psychological needs, life skills, and subjective well-being. Korean Taekwondo student athletes (*N* = 302, *M*_age_ = 17.67, range = 17–19) completed a survey evaluating basic psychological needs (autonomy, competence, and relatedness), life skills (teamwork, goal setting, social skills, time management, and leadership), and subjective well-being (life satisfaction, positive/negative affect). Data were analyzed by using descriptive statistics, correlation, and the Structural Equation Model (SEM). The model’s goodness of fit was $\varkappa^{2}$/*df* = 2.78, TLI = 0.90, CFI = 0.90, RMSEA = 0.077 (95% CI = 0.70, 0.80), and SRMR = 0.085. The three basic psychological needs were positively related to life skills and subjective well-being. In addition, life skills had a mediation effect between the three basic psychological needs and subjective well-being. The interpretation of the results indicated that life skills development and well-being depend on basic psychological needs. Thus, coaches should encourage a PYD climate to satisfy their athletes’ psychological needs.

## 1. Introduction

The theory of positive youth development (PYD) through sports has contributed to changing and developing the perspective about student athletes. Sport PYD emphasizes the psychosocial skills and inner assets acquired and developed by student athletes while participating in sports. Experts of PYD pay attention to the multidimensional skills and inner assets acquired and developed by student athletes while participating in sports [1,2]. These intrinsic assets can be developed by student athletes implicitly or explicitly through sports. Moreover, they have been studied from the perspective of “life skills”, a core concept. Life skills refer to the ability to efficiently and rationally handle various challenges and demands encountered in life, and they also refer to behavioral, emotional, and cognitive skills that contribute to a happy life [1,2,3]. Representative examples are problem-solving skills to effectively overcome unexpected adversity or problems, goal setting skills to reasonably plan life goals and directions, and communication skills to share thoughts and feelings with others. Life skills have multidimensional attributes, can be behavioral or cognitive, and include intrapersonal and interpersonal elements [4,5].

PYD researchers have conducted many studies over the past 20 years to establish the theoretical foundation of life skills [3,4,6,7]. Furthermore, they have developed tools to measure life skills [8,9], or they have developed and applied explicit programs for developing life skills and tested their effectiveness [2,10]. Additionally, PYD researchers have successfully investigated the relationships between various factors and life skills in sports. Relationships with major variables were mainly applied based on a model of coaching life skills through sport [6], a model for life skills transfer from sport to other life domains [7], and a model of positive youth development [11]. In particular, factors such as parenting attitude, social support, personality, deliberated coaching, and program quality are major factors for the development of life skills of student athletes. Although the models of previous studies search for major variables for life skills development based on the theoretical framework, they have a consistent view that basic psychological needs based on Self-Determination Theory (SDT) [12] are the starting point for life skills development.

Basic psychological needs consist of three factors: autonomy, competence, and relatedness. These three needs are directly involved in the intrinsic psychological and cognitive processes of humans and are major variables affecting actual behaviors. Autonomy means the desire to choose and act by themselves or a fundamental inclination related to an individual’s free will. Competence is the desire to prove one’s value by overcoming various challenges and demonstrating capability. Relatedness is the relief that comes from the sense of belonging and refers to the desire to belong to a social group required for achieving peace of mind [12,13].

Recent studies have argued that student athletes in an environment that guarantees the satisfaction of basic psychological needs are better at developing and transferring life skills [1,4,7]. In particular, Hodge et al. logically explain the theoretical context that is composed of the satisfaction of three basic psychological needs, the development of life skills, and psychological well-being (see Figure 1). The model is combined with SDT and Life Development Intervention (LDI). LDI is focused on self-directed change, being goal-directed, and focusing on the future, with an understanding of what needs to be achieved in the present to reach one’s best possible future [14]. This model suggests that the satisfaction level of basic needs is a major variable that determines the internalization of life skills. This argument has been supported by diverse studies. For example, Kendellen and Camiré [15] explored what kind of life skill was developed by each of the three basic psychological needs. The results showed that the satisfaction of autonomy was related to self-control and self-regulation. The results also revealed that competence needs advanced interpersonal skills, communication skills, and the ability to cope with stress. Moreover, the satisfaction of relatedness was manifested by understanding others’ emotions and social responsibility and life skills. Additionally, Cronin and Allen [16] statistically analyzed British youth sports participants and proved that the coach’s autonomy-supportive climate was an important variable in the development of life skills. The results implied that the satisfaction of autonomy was important for developing life skills. 

The ultimate goal of satisfying these basic psychological needs and developing life skills is to improve the “well-being” of student athletes. Deci and Ryan [13] reported that the satisfaction of basic psychological needs was closely related to concepts required for a happy life, such as individual self-actualization, intrinsic motivation, personal growth, life satisfaction, and positive emotions. Moreover, many researchers [1,4] have argued that developing life skills can promote athletes’ well-being. The model presented by Hodge et al. [4] logically explains that it is possible to promote the well-being of student athletes through internalization achieved by satisfying basic psychological needs and the generalization of life skills.

There are not enough studies on the role of basic psychological needs despite the logical argument that it is important for developing life skills [17]. Moreover, it is unfortunate that previous studies either focused on the supportive climate for basic psychological needs provided by the coach rather than the satisfaction level of individual student athletes’ basic psychological needs or evaluated only with autonomy among the three needs [12,18]. Especially, Hodge et al. [4] argued that it would be necessary to quantitatively evaluate the causal reasoning between basic psychological needs and life skills in their life skill development model. A few studies not only have statistically examined the structure of student athletes’ basic psychological needs, development of life skills, and well-being, but also recent studies only used a qualitative approach [15] or attempted to test the autonomy-supportive climate of the coach [16]. Therefore, it is most important to measure and evaluate the development of life skills in the sports field and the resulting well-being [3,4,6,7]. 

As mentioned above, earlier researchers have argued that the satisfaction of the three basic psychological needs is important for developing the life skills of student athletes and their well-being [1,3,4]. Particularly, the model proposed by Hodge et al. [4] clearly presents the logical structure that the satisfaction of student athletes’ basic psychological needs is a major variable that promotes the development of life skills and ultimately improves their well-being. However, although an evident theoretical model is presented, only a few studies elucidated this structure empirically. Bean and Forneris [17] claimed that the role of basic psychological needs should be statistically tested more in the research fields of positive youth development and life skills. Therefore, understanding the relationship between the satisfaction of basic psychological needs, development of life skills, and subjective well-being of student athletes in a sports environment can (a) empirically investigate the role of basic psychological needs in the field of PYD research, (b) examine the effects of the three basic psychological needs on each life skill and subjective well-being, and (c) expand the structural variables of life skills by identifying the specific roles of life skills in the relationship between basic psychological needs and well-being. It can be assumed based on the results of previous studies that there is a direct causal relationship between basic psychological needs, life skills, and subjective well-being and that life skills have a mediating effect. This study had two objectives to test these hypotheses: (a) to identify the direct causal relationship between three variables (i.e., basic psychological needs, life skills, and subjective well-being) of student athletes and (b) to understand the mediating effect of life skills on the relationship between basic psychological needs and subjective well-being. Figure 2 shows a hypothesized model of this study.

## 2. Materials and Methods 

### 2.1. Participants

The study participants were 313 Taekwondo athletes between 17 and 19 years old who were registered as athletes with the Korean Olympic Committee. This study analyzed the data of 302 participants (male = 241, female = 61, and *M*_age_ = 17.67) after excluding data from 11 participants, which were determined unreliable. They had exercised for 5.44 years (*SD* = 2.35) on average and all had competed in national competitions. Table 1 shows the specific information of study participants.

### 2.2. Measures

#### 2.2.1. Basic Psychological Needs

This study used a Korean version of the basic psychological needs scale [19]. This scale went through the internal structure and external relationship review steps to reflect the cultural characteristics of South Korea. The final scale was composed of 6 items of autonomy, 6 items of competence, and 6 items of relatedness. The items of the developed scale are measured in a 5-point Likert format (1 point = “strongly disagree”, 5 points = “strongly agree”).

Factors hindering the fitness of the model were removed after reviewing content while considering the regression coefficients and modification indices (MI) while conducting a confirmatory factor analysis for the basic psychological need scale. This study deleted No. 6 of autonomy, No. 1 of competence, and No. 2 of relatedness. The standardized regression coefficient (𝛽) for each item of three sub-factors ranged between 0.531 and 0.908. The model’s goodness of fit was 𝜘^2^/*df* = 1.812, CFI = 0.923, TLI = 0.908, RMSEA = 0.079 (90% CI = 0.059, 0.098), and SRMR = 0.083, which were within suitable levels [20]. The Cronbach’s alpha on the basic psychological need scale was 0.79, 0.85, and 0.88 for autonomy, competency, and relatedness, respectively.

#### 2.2.2. Life Skills

This study used the Korean Life Skills Scale for Sport (KLSSS) [21], a modified version of the scale developed by Cronin and Allen [8]. The KLSSS consisted of 18 items: 3 teamwork items, 4 goal setting items, 4 social skill items, 3 time management items, and 4 leadership items. Each item was measured by a 5-point Likert scale (1 point = “strongly disagree”, 5 points = “strongly agree”). In the process of the confirmatory factor analysis on life skills, social skill No. 1 and leadership No. 3 were removed. The model’s goodness of fit was 𝜘^2^/*df* = 1.804, CFI = 0.939, TLI = 0.922, RMSEA = 0.078 (90% CI = 0.075, 0.109), and SRMR = 0.063, indicating a suitable level [21]. The Cronbach’s alpha of life skill scale was 0.80, 0.91, 0.89, 0.70, and 0.78 for teamwork, goal setting, time management, social skills, and leadership, respectively.

#### 2.2.3. Subjective Well-Being

The subjective well-being scale is composed by integrating the satisfaction with life scale (SWLS) [22] and the positive and negative affect schedule (PANAS) [23]. This study used a Korean version of the satisfaction with life scale (K-SWLS) [24] and a Korean version of the positive and negative affect schedule (K-PANAS) [25]. K-SWLS has a single-factor structure composed of five items (e.g., In most ways my life is close to my ideal). K-PANAS includes 20 items: 10 items asking about positive emotions over the past two weeks (e.g., I was passionate, excited, or proud) and 10 items asking about negative emotions over the past two weeks (e.g., I was annoyed, angry, or afraid). Each item was composed on a 5-point Likert scale (1 point = “strongly disagree”, 5 points = “strongly agree”). This study deleted items 3 and 6 of positive affect and items 2 and 4 of negative affect. The model’s goodness of fit was 𝜘^2^/*df* = 2.805, CFI = 0.918, TLI = 0.908, RMSEA = 0.077 (90% CI = 0.070, 0.085), and SRMR = 0.056. The Cronbach’s alpha of the subjective well-being scale was 0.86, 0.92, and 0.92 for life satisfaction, positive affect, and negative affect, respectively.

### 2.3. Procedure

Following approval from the university’s ethics committee (IRB), participants were recruited from several high school Taekwondo teams. This study was conducted from October 2019 to May 2020. The researchers contacted team officials (usually coaches) to recruit study participants. The objectives and methods were explained to the officials in detail and consent to participate in the study was obtained from athletes. Researchers visited in person according to the schedule of each team. Just before conducting the survey, researchers explained the objective, survey method, and data processing procedures (e.g., data storage and confidentiality) of the study to the athletes once again. After that, the athletes, who voluntarily expressed their intention to participate in the study, filled out the consent form and questionnaire. The completed questionnaire was collected immediately. It was analyzed after the data cleaning process.

### 2.4. Data Analysis

The collected data were analyzed using SPSS (version 23) and AMOS (version 23) statistical programs. First, normality was tested to conduct structural equation modeling analysis. Skewness and kurtosis values were checked if they met the standard using descriptive statistics to examine whether normality assumptions were violated. It was determined that a variable met normality assumptions when the skewness value was 2 or less and the kurtosis value was 8 or less [20]. Second, Pearson correlation analysis was performed to identify the relationship between variables and multicollinearity issues. When correlation coefficient (*r*) was 0.8 or less, it was determined that there was no multicollinearity issue [20]. Third, structural equation modeling (SEM) analysis based on the maximum likelihood method was conducted to identify the relationship between variables. 𝜘^2^/*df*, CFI, TLI, RMSEA, and SRMR were used as indices for the model’s goodness of fit. The criterion of “very good fit” is 3 ≥ 𝜘^2^/*df*, 0.9 ≤ TLI, 0.9 ≤ CFI, 0.08 ≥ RMSEA, and 0.08 ≥ SRMR [21]. Fourth, bootstrapping was used to test whether the mediating effect (indirect effect) of life skills was significant. The iteration of bootstrapping was 1000 times, and the confidence interval was set at 95% using the bias-corrected method. If 0 was not within the confidence interval, the indirect effect was determined to be significant [26].

## 3. Results

### 3.1. Descriptive Statistics and Correlation

Table 2 shows the descriptive statistics and correlation coefficients of the collected data. The results of descriptive statistics revealed that all data satisfied normality assumptions. The means of the variables ranged from 2.00 to 4.29 and their standard deviations were between 0.64 and 0.91. The range of skewness was from −0.58 to 0.74 and that of kurtosis was from −0.90 to 0.00. These ranges satisfied normality assumptions [20].

The results of correlation analysis showed that three subfactors constituting basic psychological needs, five subfactors of life skills, and three subfactors of subjective well-being had significant correlations in most cases. In the correlation between basic psychological needs and life skills, the relatedness (RD) factor and the social skill (SS) factor showed the highest correlation coefficient (*r* = 0.619, *p* < 0.01), followed by the correlation between the competence (CP) factor and the leadership (LD) factor (*r* = 0.545, *p* < 0.01) and that between the CP factor and the goal setting (GS) factor (*r* = 0.526, *p* < 0.01). On the other hand, the autonomy (AU) factor and the time management (TM) factor showed the lowest correlation coefficient (*r* = 0.117, *p* < 0.05).

In the correlation between basic psychological needs and subjective well-being, the RD factor and the positive affect (PA) factor showed the highest correlation coefficient (*r* = 0.493, *p* < 0.01), followed by the correlation between the CP factor and the life satisfaction (LS) factor (*r* = 0.438, *p* < 0.01) and that between the CP factor and the PA factor (*r* = 0.411, *p* < 0.01). On the other hand, negative affect (NA) showed a significant adverse correlation with the three subfactors of basic psychological needs. Especially, it had the highest negative correlation coefficient with the AU factor (*r* = −0.247, *p* < 0.01).

Lastly, in the relationship between life skills and subjective well-being, the GS factor and the LS factor showed the highest correlation coefficient (*r* = 0.561, *p* < 0.01), followed by the correlation coefficient between the GS factor and the PA factor (*r* = 0.466, *p* < 0.01) and that between the LD factor and the LS factor (*r* = 0.448, *p* < 0.01). Meanwhile, among the subfactors of life skills, GS, SS, and LD had a significant adverse correlation with NA.

### 3.2. Structural Models

SEM analysis was carried out to test the established hypotheses and model. The items of observed variables included in PA and NA, latent variables, underwent item parceling randomly to include three observed variables each [27]. The item parceling method has the advantage of not only securing the normality of the data, but also reducing the estimation error by reducing the parameter estimate compared to methods of developing a model by using all items [27,28].

The results of SEM analysis showed that chi-square was 611.889, which was significant (*p* < 0.001). The model’s goodness of fit was 𝜘^2^/*df* = 2.78, TLI = 0.90, CFI = 0.90, RMSEA = 0.077 (95% CI = 0.70, 0.80), and SRMR = 0.085. When model fit indices were evaluated comprehensively, the SEM was judged to be good [21]. The standardized and non-standardized regression coefficients of this model are shown in Table 3. 

Paths A^1^–A^3^ represent the relationship between basic psychological needs and life skills. It was found that the three basic psychological needs statistically affected life skills. In particular, the satisfaction of relatedness needs had the greatest impact on life skills (*β* = 0.511, *p* < 0.001). The B^1^–B^3^ paths indicate the relationship between basic psychological needs and subjective well-being. Among the three basic psychological needs, autonomy (B^1^ path) and competence (B^2^ path) significantly influenced subjective well-being, but the effect of relatedness needs (B^3^ path) was not significant. The last C^1^ represents the relationship between life skills and subjective well-being. This path was significant with *β* = 0.505. The standardized regression coefficients of paths are shown in Figure 3.

### 3.3. Mediation Effect of Life Skills 

Bootstrapping was performed to test whether the mediating effect of life skills was significant in the relationship between the three basic psychological needs and subjective well-being. Table 4 shows the significance of mediating effects. The A^1^’–A^3^’ paths represent the indirect path of the three basic psychological needs (see Figure 1). As a result, it was found that the three basic psychological needs significantly affected subjective well-being through life skills. In particular, the indirect effect of relatedness was the largest (*β* = 0.258, *p* < 0.05).

## 4. Discussion

The purpose of this study was to (a) evaluate the structural relationship among the three basic psychological needs, life skills, and subjective well-being and (b) confirm the mediating effect of life skills in the relationship between the basic psychological needs and subjective well-being of student athletes. The results of this study showed that the three basic psychological needs (i.e., autonomy, competence, and relatedness) had significant relationships with life skills and subjective well-being. Moreover, the relationship between life skills and subjective well-being was also significant. Additionally, the analysis results of the mediating effect using bootstrapping revealed that life skills mediated the relationship between the three basic psychological needs and subjective well-being. These results are supported by Kendellen and Camiré [15] and Cronin and Allen [16], including the model developed by Hodge et al. [4], suggesting the theoretical structure of the three variables. Therefore, this study is meaningful in that it provides objective and empirical information to researchers who assert the theoretical relationship between basic psychological needs and life skills. Moreover, this study has contributed to the expansion of the PYD field by proving that life skills mediate the relationship between basic psychological needs and subjective well-being.

Based on the results of this study, all three basic psychological needs are important for developing the life skills of athletes and promoting their subjective well-being. However, contrary to the expectation or argument of other researchers [8,16] that autonomy needs would be most important, the influence of autonomy needs was less than the other two psychological needs. This is because how autonomy promotes life skills is applied differently depending on culture, ethnicity, and society [29]. In other words, it can be interpreted that the result is due to the unique cultural characteristics of South Korean sports. Researchers who have studied basic psychological needs have reported that it is difficult to apply or analyze autonomy in Eastern cultures [30]. This is because Eastern cultures (especially South Korea) have distinct hierarchical social and cultural structures such as seniority and social status. Such structure has an implicit norm that young or low-status people must obey and follow the instructions or control of their superiors. 

The results of this study implied that relatedness needs would play a more important role in the development of life skills and the promotion of subjective well-being than autonomy needs. Holt, Tink, Mandigo, and Fox [31] argued that the relationship between elements in the macro system (e.g., society, culture, and law) and the micro system was important for individuals to learn life skills from the perspective of human ecological theory [32]. Especially, significant others (e.g., parents, close friends, and coaches) located at the closest level are the most important factor in changing the behavior of student athletes [33]. In other words, it is believed that raising the satisfaction level of relatedness needs in interactions with them is critical for developing the life skills of athletes and improving their subjective well-being. In fact, Holt et al. [31] evaluated high school soccer teams and clearly described the effect of relatedness on life skills. Although the student athletes who were affiliated with the team had never received intentional life skills training, they had learned how to set achievable goals, how to manage time, and how to take responsibility for their actions. Holt et al. [31] interpreted these results that athletes who were affiliated with a team or group naturally learned them while interacting with each other based on their relationships.

Moreover, many studies have proved that various types of relationships experienced by athletes are important for their psychological and emotional development [34]. Particularly, the behavior of athletes can vary depending on the relational characteristics between coaches and athletes because the relationship can motivate athletes or inhibit their motivation [35]. In addition, the relatedness between coaches and players is deeply associated with the happiness of players [36]. This result indirectly indicates that the relational characteristic with the coach is an important factor in the development of athletes’ life skills and their subjective well-being. Therefore, in order to develop the life skills of athletes, it is necessary to try to make the relationship between an athlete and a person close to the athlete positive and smooth from an ecological point of view.

Meanwhile, it was confirmed that the relatedness needs in the study model affected subjective well-being indirectly through life skills, although they did not influence subjective well-being directly. In other words, the relationship among relatedness, life skills, and subjective well-being was presented using a full mediation model. The result can be interpreted that relatedness needs affect life satisfaction or emotional stability through the development of life skills. The indirect effect of relatedness was greater than that of autonomy or competence. This seems to be related to the measurement factor regarding life skills. Three of the five subfactors of life skills examined in this study (i.e., teamwork, social skills, and leadership) include the concept of social relationship or caring, and they are closely related to relatedness needs [4]. In fact, these three subfactors showed high correlation coefficients with relatedness needs. These results implied that the results of the study model could vary depending on the life skill measurement factors. Further studies are required to pay attention to understanding the specific relationship between basic psychological needs and the detailed factors of life skills.

Lastly, competence is not only involved in the acquisition and development of life skills, but also very closely related to the generalization of life skills [4,37]. In general, competence refers to the effective feeling of being able to express or exercise one’s ability in continuously generated interactions with the social environment [12]. These perceptions and feelings regarding an individual’s abilities may include skills about physical tasks (e.g., physical labor, motor functions, and sports skills) and cognitive tasks (e.g., problem-solving and decision-making) as well as social competence (e.g., interpersonal communication, etiquette, and cultural rules) [4]. Moreover, it can act as the driving force that utilizes life skills in life. Consequently, satisfying the competence needs of athletes is important not only for acquiring and developing life skills, but also for using (transferring) life skills in daily life.

The content of competence is also well described in the LDI theory [14], the theoretical foundation of the life skill development model [4]. One of the main goals of LDI is to increase the ability to practice life skills in daily life by improving the “competence” of the youth [37]. Moreover, the ultimate goal of it is to increase the chance of success in their personal goals and happiness [37]. From the perspective of LDI, competence is for the youth to have “the ability to play well, love well, think well, make relationships well, and get along well” by having a high level of self-worth [38]. As presented in this theory, having confidence in their abilities and values in sports situations greatly affects the development and generalization of athletes’ life skills. Therefore, the significant others surrounding athletes (e.g., coaches, parents, and teachers) can satisfy the competence needs of student athletes and improve the development and transfer possibility of life skills through strategies such as appropriate feedback, active support, and rewards.

In summary, the three basic psychological needs of high school Taekwondo athletes were important factors in developing life skills and improving subjective well-being. The results support the results of numerous previous studies that theoretically presented the relationship between basic psychological needs, life skills, and well-being [1,3,4,7,15]. The results also partially agree with the results of some quantitative studies [8,16] that recently attempted statistical validation. Therefore, student athletes need to satisfy three basic psychological needs to develop the life skills necessary for living in various environments (e.g., school, home, and daily life) and enjoying a happy life. The results satisfy national and social interests such as the promotion of happiness, safeguarding human rights, and the dual career of student athletes. In particular, the results of this study showing that basic psychological needs ultimately increase the well-being or happiness of players through life skills have great implications for sports PYD researchers. Future studies shall discover how to satisfy the basic psychological needs of athletes. Although some researchers have asserted that the support and satisfaction of psychological needs can promote cognitive, behavioral, and emotional changes in athletes, there is not enough discussion on when and how to intervene in detail. In other words, the next task for researchers to achieve is to prepare an efficient and practical bespoke instruction strategy based on various theoretical frameworks developed for intervening life skills and objective data [39].

## 5. Conclusions

This study examined a theoretical model consisting of basic psychological needs, life skills, and subjective well-being for high school Taekwondo athletes. The results of this study showed that (a) three basic psychological needs, life skills, and life skill utilization were significantly related and (b) life skills mediated the relationship between three basic psychological needs and subjective well-being. In conclusion, this study statistically proved that athletes with a higher satisfaction level of three basic psychological needs were more advantageous in improving life skills and subjective well-being. Therefore, significant others around athletes should use a deliberate and systematic strategy to satisfy their basic psychological needs.

Future researchers shall refer to the following information for the expansion of sports PYD. First, it is limited to generalize the results of this study to other sports because this study only evaluated Taekwondo athletes. Since the concept of basic psychological needs and life skills is highly dependent on the subject and environment, the generalization of the results to entire sports shall be performed very carefully. The environment of Taekwondo is very unique because it is an individual sport but athletes train in a group or live as a team at the same time. If a study evaluates only individual sports (e.g., golf and shooting), the results of the study can be different. In fact, this study was aimed at student athletes of all sports at the design stage, but it ended up focusing on Taekwondo athletes due to data reliability and validity issues that arose in the preliminary stage. It is believed that the data reliability and validity issues were found in the initial study design stage because of the unique characteristics of each sport. Therefore, future studies shall study these properties after classifying various sports into similar groups or by characteristics. 

Second, although this study identified that basic psychological needs affected life skills, it is still necessary to find out whether the life skills acquired and developed in the sports environment are utilized in real life. The model developed by Hodge et al. [4], the key theoretical framework in this study, showed that the ultimate purpose of life skills is to use them in everyday life, not in a sports environment. Moreover, most life skill experts [7] have argued that the true development of life skills is to utilize them in daily life. Based on these arguments, future studies are required to evaluate whether student athletes have acquired life skills in sports and to find out whether life skills are being “transferred” and utilized in actual life.

Finally, the development and transfer of life skills need to be viewed from a long-term perspective. Changes in an individual do not occur at any one moment, but they occur when one person continuously experiences factors such as a specific event, participation in a systematically designed program, and the help or support of a significant other [7]. Therefore, it is necessary to evaluate students or athletes, who participate in a program intentionally designed for developing life skills, to find out what kind of changing patterns appear from a longitudinal perspective. It will be possible to find out the time, process, and magnitude of changes specifically. The application of the latent growth model will be a good example for this objective.

## Figures and Tables

**Figure 1 ijerph-18-11538-f001:**
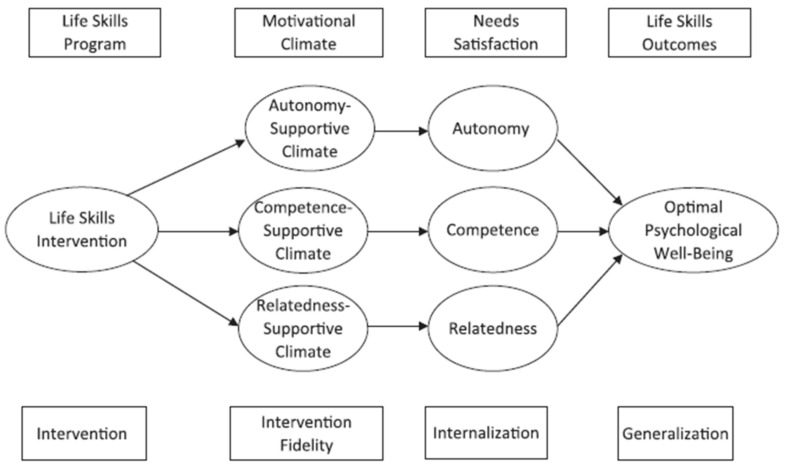
Conceptual model of life skills development (Hodge et al. 2013).

**Figure 2 ijerph-18-11538-f002:**
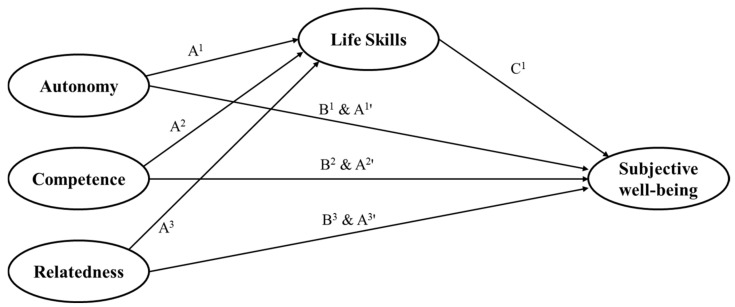
Hypothesized model of relations among basic psychological needs, life skills, and subjective well-being. Note: A^1^–A^3^ paths represent the relationship between three basic needs (autonomy, competence, and relatedness) and life skills; B^1^–B^3^ paths represent the relationship between three basic needs and SW; C^1^ path represents the relationship between life skills and SW; A^1^’–A^3^’ paths represent the mediation effect of life skills between three basic needs and SW; SW refers subjective well-being.

**Figure 3 ijerph-18-11538-f003:**
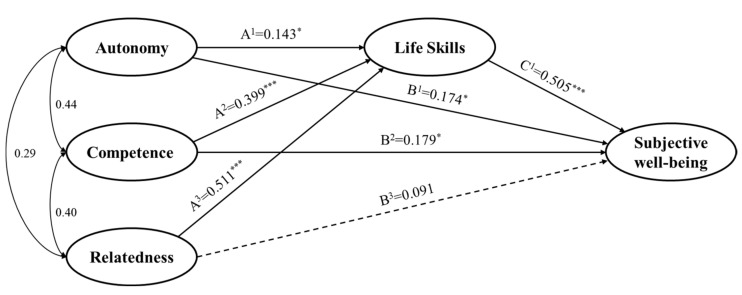
Standardized coefficients for hypothesized model (* *p* < 0.05, *** *p* < 0.001).

**Table 1 ijerph-18-11538-t001:** Characteristics of participants.

Category		*N*	%
Gender	male	241	79.8
	female	61	20.2
Age(grade)	17 (1st)	126	41.7
	18 (2nd)	151	50.0
	19 (3rd)	25	8.3
Career	1–2 yrs.	34	11.3
	3–4 yrs.	62	20.5
	5–6 yrs.	122	40.4
	more than 7 yrs.	84	27.8
Total		302	100

**Table 2 ijerph-18-11538-t002:** Descriptive statistics and correlation.

Factor	1	2	3	4	5	6	7	8	9	10	11
1. AU											
2. CP	0.415 **										
3. RD	0.319 **	0.407 **									
4. TW	0.375 **	0.355 **	0.531 **								
5. GS	0.401 **	0.526 **	0.452 **	0.534 **							
6. TM	0.117 *	0.343 **	0.216 **	0.222 **	0.479 **						
7. SS	0.353 **	0.388 **	0.619 **	0.557 **	0.484 **	0.245 **					
8. LD	0.323 **	0.545 **	0.524 **	0.538 **	0.532 **	0.421 **	0.565 **				
9. LS	0.398 **	0.438 **	0.403 **	0.325 **	0.561 **	0.406 **	0.357 **	0.448 **			
10. PA	0.340 **	0.411 **	0.493 **	0.369 **	0.466 **	0.269 **	0.378 **	0.374 **	0.553 **		
11. NA	−0.247 **	−0.241 **	−0.114 *	−0.090	−0.177 **	−0.100	−0.113 *	−0.172 **	−0.251 **	−0.176 **	
*M*	30.72	30.31	40.29	30.93	30.92	30.28	40.12	30.66	30.51	30.48	20.00
*SD*	0.73	0.80	0.64	0.70	0.73	0.91	0.70	0.77	0.83	0.81	0.84

* *p* < 0.05, ** *p* < 0.01, AU = autonomy, CP = competence, RD = relatedness, TW = teamwork, GS = goal setting, TM = time management, SS = social skill, LD = leadership, LS = life satisfaction, PA = positive affect, NA = negative affect.

**Table 3 ijerph-18-11538-t003:** Standardized and unstandardized estimates, standard error, and p-value within the models.

Path	B	*β*	S.E.	*t*
Relationship of BN and life skills				
A^1^ path (autonomy → life skills)	0.089	0.143	0.035	2.545 *
A^2^ path (competence → life skills)	0.291	0.399	0.048	6.056 ***
A^3^ path (relatedness → life skills)	0.403	0.511	0.049	8.199 ***
Relationship of BN and SW				
B^1^ path (autonomy → SW)	0.142	0.173	0.057	2.482 *
B^2^ path (competence → SW)	0.171	0.179	0.085	2.023 *
B^3^ path (relatedness → SW)	0.094	0.091	0.094	1.001
Relationship of life skills and SW				
C^1^ path (life skills → SW)	0.664	0.505	0.176	3.770 ***

* *p* < 0.05, *** *p* < 0.001, BN refers to basic needs and SW refers to subjective well-being.

**Table 4 ijerph-18-11538-t004:** Indirect standardized estimates.

Indirect Path	*β*	S.E.	95% CI	
			Lower Bounds	Upper Bounds
A^1^’ path (Autonomy → life skills → SW)	0.072 **	0.048	0.005	0.221
A^2^’ path (Competence → life skills → SW)	0.202 **	0.078	0.070	0.389
A^3^’ path (Relatedness → life skills → SW)	0.258 *	0.088	0.112	0.464

* *p* < 0.05, ** *p* < 0.01, SW refers to subjective well-being.

## Data Availability

The data presented in this study are available on request from the corresponding author. The data are not publicly available due to privacy issues.

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
