# Peer review of "Examining the Role of Life Skills in Mediating the Relationship between the Basic Psychological Needs and Subjective Well-Being of Taekwondo Student-Athletes"

_ijerph, 2021, doi:10.3390/ijerph182111538_

Round 1

Reviewer 1 Report

A very strong and relevant article. It adds some strong quantitative evidence to the literature and current knowledge base in this area. Some great knowledge claims near the end of the article as well. Discussion is strong. Methods and data analysis is sufficient as well.

There are some minor proofreading/grammar issues in the article. (e.g.  abstract line 21 "life skills development and well-being are depend on dependent on basic psychological needs". Would be good to go through and fix these.

Author Response

The researchers who are involved in this work, carefully considered your suggestions and accommodated them. As you recommend, the manuscript was changed as follows.

The title was changed as you recommended. The title of the article is ‘Examining the role of life skills in mediating the relationship between basic psychological needs and subjective well-being of Taekwondo student-athletes.

The full name of the theory was presented instead of abbreviated.

Thank you for your careful review, best regards

Reviewer 2 Report

Thank you for your submission, here you can find a document with my suggestions for it.

Round 2

Reviewer 2 Report

Thank you for your new document, the manuscript has improved considerably.